# Gut Microbiota Profile in Adults Undergoing Bariatric Surgery: A Systematic Review

**DOI:** 10.3390/nu14234979

**Published:** 2022-11-23

**Authors:** Vívian O. R. Coimbra, Louise Crovesy, Marcelo Ribeiro-Alves, Ana Luísa K. Faller, Fernanda Mattos, Eliane L. Rosado

**Affiliations:** 1Programa de Pós-Graduação em Nutrição, Instituto de Nutrição Josué de Castro, Universidade Federal do Rio de Janeiro, Avenida Carlos Chagas Filho, 373-Bloco J 2º Andar, Cidade Universitária, Rio de Janeiro 21941-902, Brazil; 2Instituto Nacional de Infectologia Evandro Chagas, Fundação Oswaldo Cruz, Avenida Brasil 4365, Rio de Janeiro 21040-360, Brazil; 3Programa de Pós-Graduação em Nutrição Clínica, Instituto de Nutrição Josué de Castro, Universidade Federal do Rio de Janeiro, Avenida Carlos Chagas Filho, 373-Bloco J 2º Andar, Cidade Universitária, Rio de Janeiro 21941-902, Brazil

**Keywords:** bariatric surgery, obesity, gut microbiota

## Abstract

Gut microbiota (GM) after bariatric surgery (BS) has been considered as a factor associated with metabolic improvements and weight loss. In this systematic review, we evaluate changes in the GM, characterized by 16S rRNA and metagenomics techniques, in obese adults who received BS. The PubMed, Scopus, Web of Science, and LILACS databases were searched. Two independent reviewers analyzed articles published in the last ten years, using Rayyan QCRI. The initial search resulted in 1275 documents, and 18 clinical trials were included after the exclusion criteria were applied. The predominance of intestinal bacteria phyla varied among studies; however, most of them reported a greater amount of *Bacteroidetes* (B), *Proteobacteria* (P), and diversity (D) after BS. *Firmicutes* (F), B, and the (F/B) ratio was inconsistent, increasing or decreasing after Roux-en-Y gastric bypass (RYGB) and sleeve gastrectomy (SG) were conducted, compared to before surgery. There was a reduction in the relative proportion of F. Moreover, a higher proportion of *Actinobacteria* (A) was observed after RYGB was conducted. However, the same was not identified when SG procedures were applied. Genera abundance and bacteria predominance varied according to the surgical procedure, with limited data regarding the impact on phyla. The present study was approved by PROSPERO, under registration number CRD42020209509.

## 1. Introduction

Obesity is a public health problem, and its prevalence has increased in recent decades; this is due, in part, to its multifactorial characteristics, which make it difficult to control [1,2,3]. It is a risk factor for the development of chronic noncommunicable diseases, such as cardiovascular, musculoskeletal, type 2 diabetes mellitus (type 2 DM), and some types of cancer, among others [1,2]. Among the recognized predisposing factors, there are genetic, environmental, and lifestyle aspects [2,3].

Recently, scientific evidence has proposed the contribution of the gut microbiota (GM) to metabolic alterations and obesity [2,3]. The GM are characterized by an aggregation of microorganisms in the gut, which are estimated, as a whole, to have one hundred times more genes than what is found in the human genome [4]. Conceptualized as a metabolic organ, they appear to play an important role in energy balance, inflammatory states, and food intake regulation [5,6]. The alteration in the GM composition has been studied as a possible cause of obesity, which may lead to an increase in the absorption of calories and the storage of body fat [7]. GM and the immune, metabolic, and neuroendocrine systems also show integrated communication, playing an important role in obesity [8].

In the face of the global obesity pandemic, bariatric surgery (BS) has been considered one of the most effective treatments for severe obesity, as well as for long-term weight reduction and maintenance. In addition, the surgical treatment has been proposed as a possible explanation in regard to the observed modifications of the GM composition after surgery [9,10,11,12]. It has been shown that BS changes both the diversity (D) and proportion of intestinal bacteria, including a decreased abundance of *Firmicutes* (F) and an increase in *Bacteroidetes* (B) and *Proteobacteria* (P) [10]. However, the impact of BS on the GM composition is varied, making it difficult to affirm the consequences of surgery and to predict the possible metabolic effects [5,13]. For this reason, we conducted a systematic review of clinical studies that analyzed GM through 16S rRNA and metagenomics techniques, thereby aiming to identify the GM characteristics of obese adults who received BS.

## 2. Materials and Methods

Search Strategy.

A systematic literature review was conducted by two independent reviewers in November 2022, using the PubMed, Scopus, Web of Science, and LILACS databases. The languages were restricted to English, Spanish, and Portuguese. The terms used for the search consisted of “bariatrics”, “gastroplasty”, “bariatric surgery”, “gastric bypass”, “jejunoileal bypass”, “stomach stapling”, “microbiot”, “microbiome”, “gastrointestinal flora”, “gut flora”, “intestinal flora”, “gastrointestinal microflora”, and “enteric bacteria”, using the Boolean operators “AND” and “OR”.

Studies that evaluated the GM profile in obese adults undergoing BS were included. Exclusion criteria were as follows: articles not published in the last ten years, not within the scope of the review, and not written in English, Portuguese, or Spanish; studies carried out in animals, pregnant women, lactating women, adolescents receiving bariatric surgery, and adults with obesity not undergoing BS; experiments with fecal microbiota transplantation, which did not assess the GM profile and without analysis of F and B; chronic noncommunicable diseases, except obesity and type 2 DM, inflammatory bowel diseases, nephropathy with the presence of *Helicobacter pylori*; intervention with probiotics, prebiotics, food supplements, and herbal medicines and medications (except in case of antidiabetic drugs).

Two researchers (V.O.R.C. and L.C.) carried out the identification and selection of the studies. They utilized the Rayyan QCRI application/website, with the intent of documenting all inclusion and exclusion decisions, allowing peer review with impartiality and traceability, thus minimizing the risk of bias [14]. After selecting studies in the databases, duplicates were eliminated. Titles and abstracts were analyzed by each reviewer, according to the exclusion criteria, and the selected articles were read in full. Data extraction occurred independently and manually, encompassing their respective methods, study designs, participant characteristics, and outcomes. Uncertainties related to inclusion and exclusion were resolved in a consensus meeting.

Outcome Measures.

The primary outcome was to verify the occurrence of alterations in the composition of the GM, analyzed by 16S rRNA and metagenomics techniques, after BS. The secondary outcome consisted of changes in anthropometric parameters, including body weight, body mass index (BMI), and the remission of obesity-related diseases, such as type 2 DM. The main aspects of interest for article selection are described in Table 1.

The present study was approved by the public database of protocols for systematic reviews with health outcomes PROSPERO, under registration number CRD42020209509.

## 3. Results

The applied search strategy returned a total of 1275 published articles, 8 in LILACS, 432 in PubMed, 555 in Scopus, and 280 in Web of Science, between November 2012 and November 2022, of which 518 were duplicates. After screening by title and abstract, as well as the full text when necessary, 18 studies were included in the systematic review, as shown in Figure 1.

Relevant data from the studies included in this systematic review are summarized in Table 2.

The studies added to the systematic review and the results of interest are shown in Table 3 and Table 4.

Of the selected studies, 15 out of 18 (83 %) were conducted after an RYGB procedure [5,15,16,17,18,19,20,21,22,23,25,26,28,30,31]. Of those, eight included both male and female populations [15,16,17,19,22,25,28,30,31], four included only women [18,20,23,27], and two did not report sex [5,26]. The SG procedure appeared in 12 of 18 studies [5,15,16,17,19,20,23,25,27,29,30]; the majority included both men and women (eight studies) [15,16,17,19,24,25,27,30], three recruited only females [20,23,29], and one did not provide the sex of the population [5]. The postoperative follow-up time of the studies ranged from one month to eight years, including 1 [29], 3 [25], 3.4 (0.9–9.6) [23], 6 [19,20,27,28,31], 9.60 ± 3.92 [16], and 12 months [5,17,21,26,30], as well as longer periods of 4 [15], 5 [18] and 8.3 ± 1.7 years [22].

## 4. Discussion

The interaction between GM and BS is complex since surgery itself results in anatomical and physiological changes in the intestine. It is a multifaceted condition, where in addition to the surgical modifications, food consumption is altered, and weight loss occurs quickly after surgery, conditions that impact the GM. On the other hand, the GM composition seems to influence the prognosis of weight loss and metabolic improvement [5,10,20,32]. In addition to intestinal bacteria, microbial metabolites appear to play an important role in the physiological and health changes regardless of the surgical procedure [33,34]. Metabolites derived from microbial metabolism, including short-chain fatty acids, secondary bile acids, betaine and choline, may act synergistically and beneficially in human metabolism and BMI reduction after BS [34,35]. In a longitudinal study with severely obese adults undergoing RYGB or SG, significant changes in the GM composition and microbial metabolites were observed between the pre- and postoperative periods [35]. Furthermore, Juárez-Fernández et al. observed a significant reduction in the concentrations of acetate, butyrate, and propionate after BS [15].

Modifications in the GM after BS have been associated with improved glucose homeostasis, weight loss, changes in food course and motility in the gastrointestinal tract, and changes in nutritional status and diet therapy after BS [6,10,26]. The necessary changes in food intake after surgery, resulting in an energy-restricted and high-protein diet, in addition to a supplementation protocol, impact food digestion and absorption as well as the GM composition [10].

Murphy et al. observed a reduction in BMI and type 2 DM remission after one year of both SG and RYGB [30]. Koffer et al. observed type 2 DM remission after six months of BS in 80% of the population with the disease, suggesting that weight loss and reduction in insulin resistance were related [20]. In those individuals that presented type 2 DM remission, there was a significant increase in the genus *Roseburia intestinalis*, from phylum F. This increase was also described in other recent studies, regardless of the surgical procedure, associated with a beneficial effect on improved insulin sensitivity, corroborating the hypothesis that alterations in the composition of the GM after BS may be associated with remission of DM. It should be noted, however, that changes in the proportion of phylum F after BS were still heterogeneous in both surgical procedures [17,23,30].

In obese individuals, GM dysbiosis has been documented, especially towards a greater relative abundance of F and a reduction in B and D, with modifications regarding the quantity and variability of bacterial species. Most studies in the present review corroborated the indication that D decreased with BS. Studies that showed an increase in F, associated this modification with the higher energy and fatty acids uptake and BMI [32].

The literature has shown that a lower F/B ratio is associated with weight loss and metabolic improvement [21]. However, the studies included in this review were contradictory on this topic, regardless of the surgical procedure and the postoperative period analyzed.

The increase in P abundance, observed in different postoperative periods of RYGB and after six months of SG, may be due to greater transient oxygen exposure and changes in the gut pH as a result of BS [32]. In mice submitted to BS, a higher P abundance was related to improved insulin sensitivity, suggesting a beneficial role of this phylum in glucose metabolism [23].

The relative abundance of the genus *Veillonella*, from the F phylum, was higher in only four of the sixteen studies with RYGB, and the same was not observed in the SG procedure [16,19,21,25]. This bacterium is found in the mouth tract and may have its abundance exacerbated in RYGB due to reduced exposure to the acidic compartment of the stomach, providing aerotolerant colonization and favoring the access of oral bacteria in the intestine [19].

In patients undergoing RYGB, a negative correlation was observed between the BMI and five genera of bacteria, including *Veillonella*. The relative abundance of this bacteria was higher after three months of BS, when compared to the preoperative period, and associated with BMI reduction. The higher proportion of *Veillonella* may be due to anatomical modifications on stomach size and the oral microbiota composition after surgical intervention and has been linked to the control of inflammation and body weight [27].

*Akkermancia muciniphila*, from the phylum *Verrucomicrobia*, has been considered to have an anti-obesity effect and enhance type 2 DM remission [36]. This bacterial genus had a high relative abundance in four of the seventeen experiments with RYGB [16,18,23,26] and in three of the nine studies with SG [5,25,26]. However, a decrease was observed in three participants undergoing RYGB. This bacterium appears to be associated with the modulation of the immune response and the homeostasis of the basal metabolism in germ-free mice and with weight loss and metabolic control after BS [26].

As for *Streptococcus*, the genus of phylum F, had greater abundance in only two of the thirteen studies with RYGB and in one of the nine studies with SG, which may show the survival and proliferation of aerotolerant bacteria [19,21,27]. A study with a European metagenome found the significant growth of *Streptococcus* in patients with persistent type 2 DM one year after the surgical procedure, suggesting a positive association between the expansion of this genus of bacteria and the risk of this chronic disease [30].

*Faecalibacterium prausnitzii*, despite evidence associating its abundance with reduced plasma glucose levels and increased insulin sensitivity and possible anti-inflammatory effect [23,37], showed contrasting results after BS for both surgeries [19,23].

In general, RYGB surgery seemed to result in a major modification of the GM composition compared to SG [19,31]. Thus, although both procedures of BS result in similar dietary recommendations and postoperative food intake and promote weight loss and the remission of type 2 DM in obese patients, RYGB appears to lead to functional changes in the GM, including intestinal motility, changes in bile acid flow, and intestinal hormones [5,10]. The acid–base balance and pH regulation are important for an adequate immune response in these patients [3]. After BS, reduced gastric volume can elevate the pH and oxygen levels in the stomach and distal intestine, allowing the inhibition of anaerobic microorganisms and the proliferation of facultative aerobics, including P, *Akkermansia muciniphila*, *Escherichia coli*, *Bacteroides* spp., and bacteria associated with the oral microbiota [10], as observed in this systematic review.

GM appears to stimulate the immune system and the enteric nervous system, modulating the central nervous system and possibly impacting the hypothalamic signaling of hormones related to hunger and satiety, immune regulation, intestinal motility and secretion, and intestinal mucosal homeostasis. This mechanism of interaction between the GM, the immune system, and the neuroendocrine system has been associated with intestinal permeability, inflammatory state, changes in feeding behavior, and bacterial survival and growth [7], which could explain, in part, the importance of GM in the surgical prognosis.

The heterogeneity of data on the impact of BS on the GM, is partly due to the small sample sizes, the lack of information and/or control of dietary intake and gastric pouch size after surgery, studies with only one sex or no information regarding the sex of the study population, and the lack of information on the presence of diseases associated with obesity [5,14,22,25,30]. Other variables that can lead to bias in the studies described are hospitalization alone, changes in diet, food preference and consistency, an inadequate diet after surgery, the use of medications (for different prophylaxes to eradicate *Helicobacter pylori* or urinary tract infection, for example), the use of antibiotics in the perioperative phase and supplements, complications after BS, withdrawal of participants during the research, and the use of different surgical procedures and procedures for DNA extraction for analysis of the GM composition [16,17,31]. Furthermore, a specific limitation of this study was the exclusion of 23 articles that did not analyze the F/B ratio, which could have led to selection bias.

The long-term impact of BS on the GM is not yet known, particularly in terms of postoperative follow-up greater than one year, with most studies having up to six months [19,20,23,27,28,29,31]. Due to multiple interfering factors resulting in possible biases, conclusions on the effect of BS on the GM and vice versa should be evaluated with caution.

## 5. Conclusions

Obesity surgical treatment, such as BS, has a positive impact on lipid and glucose metabolism, remission of type 2 DM, and weight loss and also results in GM changes. In patients undergoing RYGB, an increase in B, *Actinobacteria* (A), P, and D was observed in most studies with no consistency regarding the F/B ratio. After SG, there was an increase in the proportion of B, P, and diversity, with no reports on A or consensus on the F/B ratio. In both surgical procedures, there were reports of a decreased proportion of F. For specific bacteria genera, the literature available is not necessarily the same as for phyla. The magnitude of the modifications on the abundance of bacteria is also unknown.

The results are controversial, differ according to the surgical procedure, and may change depending on the postoperative period studied; thus, it is not possible to state whether changes in the GM would be permanent. Additionally, the literature available cannot discriminate between whether the GM changes are due to the BS itself (hormonal, anatomical, intestinal functional, and microbiological) and not to the diet and lifestyle modifications that also occur after surgery, for example. For now, it is not prudent to state the magnitude of the influence of changes to the GM, as a contributing factor for weight loss promotion and metabolic improvement after BS.

## Figures and Tables

**Figure 1 nutrients-14-04979-f001:**
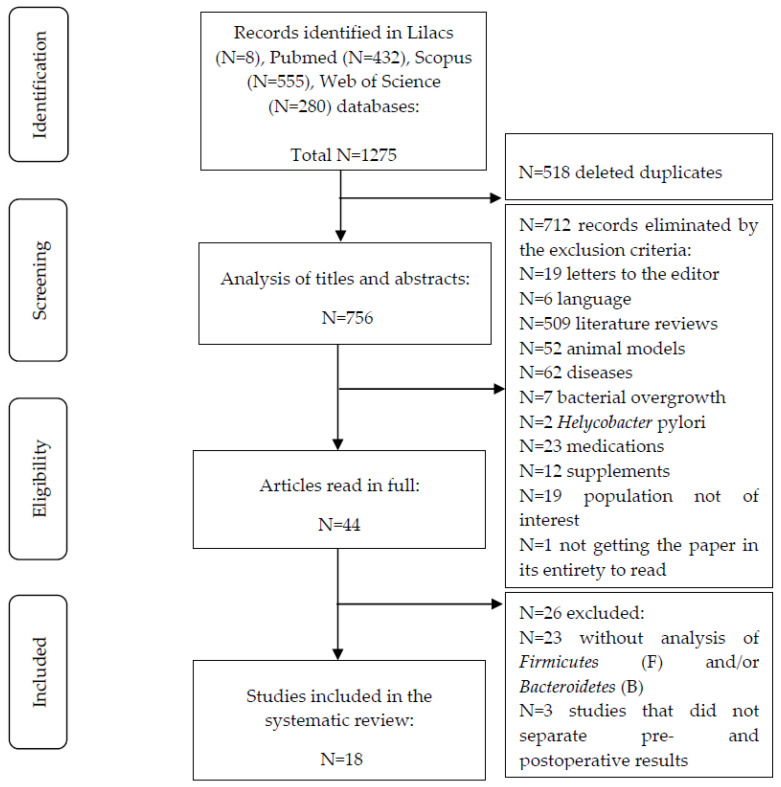
Flowchart of the study.

**Table 1 nutrients-14-04979-t001:** Aspects of interest for the initial selection of articles.

Parameters	Defined Criteria
Population	Individuals over the age of 18 with obesity or who were overweight.
Intervention	Bariatric surgery: sleeve gastrectomy (SG) and Roux-en-Y gastric bypass.
Comparison	Comparison of the gut microbiota profile at different pre- and postsurgical stages.
Outcomes	Identification of the impact of the BS on the composition of the GM.
Designs	Cohort studies, prospective longitudinal, nonrandomized, randomized clinical trial, and randomized controlled clinical trials.

**Table 2 nutrients-14-04979-t002:** Summary of reviewed studies.

Authors, Country	Study Population at Baseline (Age, BMI)	Sample Size (Surgical Procedures, Sex)	Study Design	Sequencing/Genetic Analysis	Stool Collection Period	Time of Followup
Juaréz-Fernandes et al., 2021 [15], Spain	Age (years): 18–60 BMI (kg/m²): 45.46 ± 2.05	(N = 9) RYGB: (N = 1) SG: (N = 6) BPD: (N = 2) (M:F): 2:7	Longitudinal	16S rRNA (V3–V4) gene sequencing	Before and four years after BS	4 years
Chen et al., 2020 [16], China	Age (years): 30.92 ± 9.17 RYGB: 33.24 ± 10.13 SG: 29.50 ± 8.31 BMI (kg/m²): 40.84 ± 10.67 RYGB: 45.75 ± 14.26 SG: 37.84 ± 6.16	(N = 87) RYGB: (N = 33) (M:F): 14:19 SG: (N = 54) (M:F): 13:41	Longitudinal	16S rDNA (V3-V4) sequencing, RT-PCR	Before and 3 months after BS	9.60 ± 3.92 months
Davies et al., 2020 [17], New Zealand	Age (years): 20–56 RYGB *: 48.5 ± 5.5 SG *: 47.7 ± 6.9 BMI (kg/m²): 35–65 RYGB *: 38.2 ± 5.7 SG *: 40.0 ± 5.9	(N = 44) RYGB: (N = 22) (M:F): 7:15 SG: (N = 22) (M:F): 14:8	Randomized Controlled Trial	Genome shotgun sequencing	2 days before and 1 year after BS	12 months
Faria et al., 2020 [18], Brazil	Age (years): 18–65 BMI (kg/m²): 35–49.9	(N = 34) CG (preoperative patients): (N = 8) F: 8 RYGB: (N = 26 ) Non-regain: (N = 12) Regain: (N = 14) F: 26	Cross-sectional	16S rRNA gene sequencing (V3–V4)	RYGB non-regain: before and 55 months after BS RYGB regain: before and 84 months after BS	At least 5 years RYGB non-regain *: 54.9 ± 34.5 months RYGB regain *: 83.8 ± 40.8 months
Farin et al., 2020 [19], France	Age (years): ≥18 BMI (kg/m²): ≥35	(N = 197) RYGB: (N = 89) SG: (N = 108) Both sexes	Cohort	Shotgun metagenomic sequencing	1 month before and 6 months after BS	6 months
Koffert et al., 2020 [20], Finland	Age (years): 18–60 BMI (kg/m²): ≥35 40.9 ± 4.2	(N = 27) RYGB: (N = 6) SG: (N = 7) Controls: (N = 14) F:27	Clinical trial	16S rRNA gene sequences	Before and 6 months after BS	6 months
Al Assal et al., 2019 [21], Brazil	Age (years): 18–60 RYGB *: 45.80 ± 7.95 BMI (kg/m²): ≥35 RYGB *: 46.40 ± 5.48	(N = 25) RYGB: (N = 25) F: 25	Cohort	16S rRNA gene sequencing (V4)	Before and 3 and 12 months after BS	12 months
Gutiérrez-Repiso et al., 2019 [22], Spain	Age (years): ≥18 RYGB *: 43.33 ± 9.97 BMI* (kg/m²): 47.03 ± 6.01	(N = 24) RYGB: (N = 24) Both sexes	Prospective cohort	16S rRNA (V2, 3, 4, 6-7, 8, and 9) metagenomic sequencing	Before and 8.3 ± 1.7 * years after BS	8.3 ± 1.7 * years
Lee et al., 2019 [23], USA	Age ** (years): 52.5 (32–62) RYGB **: 57 (43–60) SG **: 45 (41–53) BMI (kg/m²): 30–40 RYGB **: 35.1 (31.3–38.6) SG **: 35.8 (33.0–37.6)	(N = 12) MWL: (N = 4) RYGB: (N = 4) SG: (N = 4) F: 12	Randomized controlled pilot trial	16S rRNA (V3–V4) amplicon sequencing	RYGB: Before and 1.8 (0.9–5.6) ** after BS SG: Before and 2.3 (2.1–4.3) ** after BS	3.4 (0.9–9.6) ** months RYGB **: 1.8 (0.9–5.6) SG**: 2.3 (2.1–4.3)
Lin et al., 2019 [24], USA	Age (years): 20–64 SG *: 36.2 ± 9.9 BMI (kg/m²): ≥30 SG*: 35.9 ± 4.0	(N = 10) SG: (N = 10) (M:F): 4:6	Longitudinal	16S rRNA (V4) amplicon sequencing	Before and 1 and 3 months after BS	3 months
Sánchez-Alcoholado et al., 2019 [25], Spain	Age (years): 26–63 BMI (kg/m²): RYGB: 43.7 ± 5.3 SG: 46.9 ± 6.6	(N = 28) RYGB: (N = 14) (M:F): 4:10 SG: (N = 14) (M:F): 4:10	Longitudinal	16S rDNA genes next-generation sequencing	Before and 3 months after BS	3 months
Cortez et al., 2018 [26], Brazil	Age (years): 18–64 DJBm *: 47 ± 8 BMI (kg/m²): 25.0–39.9 DJBm *: 29.7 ± 1.9	(N = 21) Standard medical treatment: (N = 10) DJBm: (N = 11) Sex: not stated	Randomized controlled trial	16S rRNA (V4) gene sequencing	Before and after 6 and 12 months	12 months
Kikuchi et al., 2018 [27], Japan	Age (years): 18–65 LSG-DJB *: 48.0 ± 2.5 SG *: 40.7 ± 2.0 BMI (kg/m²): >30	(N = 44) LSG-DJB: (N = 18) (M:F): 10:8 SG: (N = 22) (M:F): 11:11 LAGB: (N = 4) (M:F): 0:4	Nonrandomized prospective observational clinical trial	16S rDNA sequencing, RT-PCR	1, 3 and 6 months	6 months
Chen et al., 2017 [28], China	Age * (years): 51.5 ± 9.6 BMI (kg/m²): ≥40 RYGB *: 46.3 ± 4.7	(N = 24) RYGB: (N = 24) (M:F): 14:10	Cohort	16S rDNA sequencing, RT-PCR	Before and 180 days after BS	6 months
Medina et al., 2017 [5], Chile	Age (years): 18–60 BMI (kg/m²): 30–50 RYGB *: 37.1 ± 2.8 SG *: 35.2 ± 2.4	(N = 19) MD: (N = 9) RYGB: (N = 5) SG: (N = 5) Sex: not stated	Cohort	16S rRNA gene sequencing (V3–V4), RT-PCR	Before and 6 months after BBS	12 months
Sanmiguel et al., 2017 [29], EUA	Age * (years): 39.5 ± 8.7 BMI * (kg/m²): 44.1 ± 5.6	(N = 8) SG: (N = 8) F: 8	Longitudinal	16S rRNA gene sequencing (V4)	Before and 1 month after BS	1 month
Murphy et al., 2016 [30], New Zealand	Age (years): RYGB *: 48.6 ± 6.1 SG *: 48.3 ± 6.1 BMI (kg/m²): RYGB *: 38.4 ± 5.2 SG *: 36.9 ± 5.1	(N = 14) RYGB: (N = 7) (M:F): 3:4 SG: (N = 7) (M:F): 5:2	Double-blind clinical trial	Shotgun metagenomic sequencing	Before and 1 year after BS	12 months
Ward et al., 2014 [31], USA	Age (years): 18–70 BMI (kg/m²): ≥40 RYGB *: 47.1 ± 4.8	(N = 8) RYGB: (N = 8) (M:F): 1:7	Longitudinal	16S rRNA gene sequencing (V4)	1 month before and 6 months after BS	6 months

Results were expressed as mean ± SD * or median (range) **. BMI, body mass index; BS, bariatric surgery; DJBm, duodenal-jejunal bypass surgery with minimal gastric resection; BPD, biliopancreatic diversion; F, female; LAGB, laparoscopic adjustable gastric banding; LSG-DJB, laparoscopic sleeve gastrectomy with duodenojejunal bypass; M, male; MD, medical dietary treatment; MWL, medical weight loss; R, ribosomal; RT-PCR, reverse transcription polymerase chain reaction; RYGB, Roux-en-Y gastric bypass; SG, sleeve gastrectomy; USA, United States of America.

**Table 3 nutrients-14-04979-t003:** Comparison of the *Bacteroidetes*, *Firmicutes*, *Firmicutes* and *Bacteroidetes* ratio, and specific bacteria between the RYGB and SG surgeries.

Surgical Procedures	*Bacteroidetes*	*Firmicutes*	*Firmicutes* and *Bacteroidetes* Ratio	Specific Bacteria
RYGB	Increased: 6 months [5,26,28]; 12 months [17,26]. Decreased: 3 months [16]; 6 months [20]; 5–7 years [18].	Increased: 12 months [17,30]. Stable: 3 months [16]. Decreased: 6 months [5,19,26]; 4 years [15].	Decreased: 6 months [5].	B: Increased in 6 months for *Succiniclastum* sp., *Bacteroides*, *Bacteroides coprophilus*, *Bacteroides eggerthii* [5], *Bacteroides*, *Alistipes* [20,26]. F: Increased in 6 months for *Clostridiaceae*, *Clostridium*, *Veillonella*, *Granucatiella*, *Oscillospira* [25], *Streptococcus* [20,21], *Sporobacter termitidis* [20], *Veillonella* [21], *Gemella*, *Granulicatella* [16], *Lactobacillus*, *Enterococcus* [28], *Lactobacillales* sp. [5], *Dialister*, *Ruminococcus*, *Roseburia*, *Acidamicoccus* [25], *Streptococcus*, *Veillonella*, *Roseburia*, *Enterococcus faecalis* [19]; in 9 months for *Faecalibacterium prausnitzii* [23]; in 4 years for *Clostridiaceae* [14]; in 5–7 years for *Streptococcus*, *Enterococcus*, *Lachnobacterium* [18]. Decreased in 3 months for *Peptostreptococcaceae* [25]; in 4 years for *Coprococcus Acinetobacter, Coprococcus, Lachnospira, Lactococcus, Megamonas, Oribacterium, Phascolarctobacterium* [14]; in 5–7 years for *Faecalibacterium* [18].
SG	Increased: 1 and 3 months [27]; 12 months [17,29]. Decreased: 6 months [5,20].	Increased: 6 months [5]. Stable: 3 months [16]. Decreased: 6 months [19]; 12 months [29]; 4 years [15].	Trend of Increase: 1 and 3 months [27]. Increased: 6 months [5]. Decreased: 12 months [29].	B: Decreased in 3 months for *Butyricimonas* [16]. Increased in 6 months for *Alistipes* [20]. F: Increased in 1 and 3 months for *Streptococcus* [27]; in 3 months for *Gemella*, *Granulicatella*, *Faecalibacterium* [16]; in 6 months for *Streptococcus luteciae* [5], *Streptococcus* spp. [20], *Sporobacter termitidis* [20], *Clostridium*, *Anaerostipes hadrus*, *Flavonifractor plautii*, *Ruminococcus gnavus*, *Oscillibacter* sp. *KLE*, *Veillonela*, *Streptococcus* [19]; in 12 months for *Roseburia intestinalis*, *Streptococcus*, *Lactobacillus* [30], *Bulleidia* [29]; in 4 years for *Clostridiaceae, Acinetobacter, Coprococcus, Lachnospira, Lactococcus, Megamonas, Oribacterium, Phascolarctobacterium* [15]. Decreased in 3 months for *Clostridiaceae*, *Anaerostipes* [25]; in 6 months for *Ruminococcus gnavus*, *Faecalibacterium prausnitzii* [19]; in 4 years for *Coprococcus* [15].

B, *Bacteroidetes*; BS, bariatric surgery, F, *Firmicutes;* F/B, *Firmicutes/Bacteroidetes*; RYGB, Roux-en-Y gastric bypass; SG, sleeve gastrectomy.

**Table 4 nutrients-14-04979-t004:** Comparison of *Actinobacteria*, *Proteobacteria,* diversity, and specific bacteria between the RYGB and SG surgeries.

Surgical Procedures	*Actinobacteria*	*Proteobacteria*	Diversity	Specific Bacteria
RYGB	Increased: 6 months [5]; 9 months [23]; 12 months [30].	Increased: 6 months [5]; 9 months [23]; 12 months [17]; 4 years [15]; 5–7 years [18].	Trend of increase: 9 months [23]; 12 months [21]. Increased: 3 months [16]; 6 months [14,19,26]; 12 months [26,30]; 4 years [15]; 5–7 years [18]. Stable before and after BS: 3 months [25]; 6 months [31]; 12 months [17]. Decreased: 8,3 ± 1,7 years [22].	A: Increased in 6 months for *Bifidobacterium* [28]; in 3 months for *Slackia*. Decreased in 3 months for *Bifidobacteriaceae*, *Bifidobacterium*, *Collinsella* [25]; in 6 months for *Bifidobacteria bifidum* [19]. P: Increased in 3 months for *Enterobacteriacea* [25], *Neisseria* [21], *Klebsiella*, *Haemophilus* [16]; in 6 months for *Citrobacter* [5]; in 12 months for *Enterobacteriales* [17], *Escherichia coli*, *Klebsiella pneumoniae*, *Haemophilus parainfluenzae* [19]; in 4 years for *Enterobacteriaceae*, *Sinobacteriaceae* [15]; in 5–7 years for *Succinivibrio*, *Klebsiella* [18]. Decreased in 6 months for *Escherichia* [28]; in 4 years for *Acinetobacter* [15]. *Verrucomicrobia (Akkermansia muciniphila):* Increased in median 1.75 months [23]; in 6 and 12 months [26]; in 9.60 ± 3.92 months [16]; in non-regain group in 5 years. Stable in regain group (15% weight gain increase after the lowest weight after BS) in 5 years [18].
SG	_	Increased: 6 months [5]; 4 years [15].	Increased: 3 months [16,24]; 6 months [19,20]; 4 years [15]. Stable before and after BS: 12 months [17]. Stable between RYGB and Sleeve: 3 months [25].	A: Increased in 12 months for *Atopobium* [29]. Decreased in 3 months for *Bifidobacteriaceae*, *Bifidobacterium* [25], *Actinomyces* [16]; in 6 months for *Bifidobacteria dentium* [19]; in 12 months for *Bifidobacteriaceae* [29]. P: Increased in 3 months for *Haemophilus*, *Klebsiella* [16]; in 6 months for *Enterobacteriales Bulleidia*, *Escherichia coli* [5], *Klebsiella pneumoniae, Haemophilus parainfluenzae* [19]; in 4 years for *Enterobacteriaceae*, *Sinobacteriaceae* [14]. Decreased in 3 months for *Oxalobacter*, *Sutterella*, *Desulfovibrio* [16]; in 4 years for *Acinetobacter* [14]. *Verrucomicrobia (Akkermansia muciniphila):* Increased in 3 months [27]; in 6 months [5]; in 9.60 ± 3.92 months [16].

A, *Actinobacteria*; BS, bariatric surgery; P, *Proteobacteria;* RYGB, Roux-en-Y gastric bypass; SG, sleeve gastrectomy.

## Data Availability

Not applicable.

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
