# Peer review of "Gut Microbiota Profile in Adults Undergoing Bariatric Surgery: A Systematic Review"

_nutrients, 2022, doi:10.3390/nu14234979_

Round 1

Reviewer 1 Report

Coimbra et al did a systematic review of papers examining changes pre and post bariatric surgery. Overall the paper is well structured and shows that the current data is conflicting due to multiple cofounders. My only comment are minor and can be addressed. the Authors summarize the work based on the 4 largest phyla: Bacteroidetes, Firmicutes, Actinobacteria, and Proteobacteria. 

1. . They excluded 26 papers that did not analyze F/B ratio and so their discussion on F/B ratio may be biased due to selection bias. In their discussion, they should mention that this is a specific limitation.

2. Furthermore, the authors should mention the total number of patients that were included in their 16 trials separated by surgery. Additionally, they should create another table listing out the 16 studies included in their review, what type of sequencing was done (i.e. 16s vs shotgun), time of follow up, when stool collection was done etc. 

3. the authors should add to table 3, verrucomicrobia as this phylum includes Akkermansia and Akkermansia is quite important in weight loss and obesity.  

Author Response

Response to Reviewer 1 Comments

Coimbra et al did a systematic review of papers examining changes pre and post bariatric surgery. Overall the paper is well structured and shows that the current data is conflicting due to multiple cofounders. My only comment are minor and can be addressed. the Authors summarize the work based on the 4 largest phyla: Bacteroidetes, Firmicutes, Actinobacteria, and Proteobacteria.

Point 1: They excluded 26 papers that did not analyze F/B ratio and so their discussion on F/B ratio may be biased due to selection bias. In their discussion, they should mention that this is a specific limitation.

Response 1: We would like to thank you sincerely for receiving our manuscript and for your valuable comments. Your suggestion has been included in the paper. In the discussion, we mentioned the exclusion of articles that did not analyze F/B ratio enzatizing that it is a specific limitation. We emphasize that the literature search was redone in November 2022 for updating and the number of articles excluded for this reason was 23.

Point 2: Furthermore, the authors should mention the total number of patients that were included in their 16 trials separated by surgery. Additionally, they should create another table listing out the 16 studies included in their review, what type of sequencing was done (i.e. 16s vs shotgun), time of follow up, when stool collection was done etc.

Response 2: Thank you for the recommendation and for the opportunity to improve the work. The total number of patients that were included in the paper were mentioned, according to the surgery in the Table 2, in column Sample Size (surgical procedures, gender). Furthermore, accepting your important suggestion, a Table 2 was included with the summarized of relevant data from the 16 studies included in this systematic review: Authors, country; Study Population in baseline (age, BMI); Sample Size (surgical procedures, gender); Study Design; Sequencing/Genetic Analysis; Stool Collection Period and Time of Follow-up.

Point 3: The authors should add to table 3, verrucomicrobia as this phylum includes Akkermansia and Akkermansia is quite important in weight loss and obesity.

Response 3: Thank you for your sugestion, relevant information of Verrucomicrobia (Akkermansia muciniphila) were included in the table: Comparison of Actinobacteria, Proteobacteria, diversity and specific bacteria between RYGB and SG surgeries.

Reviewer 2 Report

This systemic review of literature of changes in gut microbiota following the two most common weight loss/metabolic surgical procedures is a very timely piece. The paper is overall well-written and the inclusion of literature in Portuguese and Spanish, as well as the succinct, intelligent discussion represent special strength.

The reviewer's major concern is the narrow time-period of reviewed literature ending in October 2020. This is a weakness greatly limiting the usefulness of the information to the field, and therefore, it is recommended the authors update the review to include references/studies for the past two years.

Minor recommendations:

Since there is no discussion on 'age' as a contributing factor or the age-range specified, it appears to be not important to emphasize that the review is on adults AND elderly people. Also, the exclusion criteria should state that adolescent receiving bariatric surgery were excluded.

Please, consider using the term 'Sleeve Gastrectomy (SG)' instead 'sleeve'.

You may also want to refer to RYGB and SG as 'procedures' or 'surgeries' rather than as 'techniques'.

Author Response

Response to Reviewer 2 Comments

This systemic review of literature of changes in gut microbiota following the two most common weight loss/metabolic surgical procedures is a very timely piece. The paper is overall well-written and the inclusion of literature in Portuguese and Spanish, as well as the succinct, intelligent discussion represent special strength.

Point 1: The reviewer's major concern is the narrow time-period of reviewed literature ending in October 2020. This is a weakness greatly limiting the usefulness of the information to the field, and therefore, it is recommended the authors update the review to include references/studies for the past two years.

Response 1: I would like to thank you sincerely for receiving our manuscript and for your valuable recommendations. We understand the issue and, to update, the entire literature search was carefully redone in November 2022, including all references/studies published from 2020 onwards.

Point 2: Since there is no discussion on 'age' as a contributing factor or the age-range specified, it appears to be not important to emphasize that the review is on adults AND elderly people. Also, the exclusion criteria should state that adolescent receiving bariatric surgery were excluded.

Response 2: Thanks for the observation, we remove the term “AND elderly people”, and included in the exclusion criteria that adolescent receiving bariatric surgery were excluded. 

Point 3: Please, consider using the term 'Sleeve Gastrectomy (SG)' instead 'sleeve'.

Response 3:  Thanks, the correct term 'Sleeve Gastrectomy (SG)', instead 'sleeve', was included in the entire study.

Point 4: You may also want to refer to RYGB and SG as 'procedures' or 'surgeries' rather than as 'techniques'.

Response 4: Grateful for your recommendation, we refer RYGB and SG as 'procedures' or 'surgeries' rather than as 'techniques' throughout the text.

Reviewer 3 Report

In the text of table 1 I propose to remove "(BS), (RYGB) and (GM)", both because such acronyms are given in the table legend and previously in the text.

Author Response

Response to Reviewer 3 Comments

Point 1: In the text of table 1 I propose to remove "(BS), (RYGB) and (GM)", both because such acronyms are given in the table legend and previously in the text.

Response 1: I would like to thank you sincerely for receiving our manuscript and for your valuable propose. In the text of table 1, we remove "(BS), (RYGB) and (GM)". Sleeve Gastrectomy (SG) was included, because was the first time it appeared in the text.

Round 2

Reviewer 2 Report

The authors have been very receptive to my critiques and addressed all my comments. The inclusion of studies in the review from the past two years improved greatly the manuscript, and in turn, its potential impact on the field. My only recommendation is to check the quality of Figure 1 before publication.

Author Response

We would like to thank you sincerely for receiving our manuscript and for your valuable comment. Your suggestion has been included in the paper, the Figure 1 has been included in better resolution.

Futhermore, one study in the systematic review was excluded because the authors had not informed the age of the participants.  This morning, the authors answered the e-mail, and the study was included in the systematic review, totaling 18 studies included. I have already sento an e-mail to Nutrients informing them and we are forwarding the update version as attachment.